# Hydrogel Small-Diameter Vascular Graft Reinforced with a Braided Fiber Strut with Improved Mechanical Properties

**DOI:** 10.3390/polym11050810

**Published:** 2019-05-06

**Authors:** Guoping Guan, Chenglong Yu, Meiyi Xing, Yufen Wu, Xingyou Hu, Hongjun Wang, Lu Wang

**Affiliations:** 1Engineering Research Center of Technical Textiles, Ministry of Education; Key laboratory of Textile Science and Technology, Ministry of Education; College of Textiles, Donghua University, Songjiang District, Shanghai 201620, China; ggp@dhu.edu.cn (G.G.); Belongdrew@163.com (C.Y.); 18363994765@163.com (M.X.); yufenwu1130@126.com (Y.W.); huxingyou@126.com (X.H.); 2Department of Biomedical Engineering, Stevens Institute of Technology, Hoboken, NJ 07030, USA; Hongjun.Wang@stevens.edu

**Keywords:** hydrogel, vascular graft, braided fiber strut, swellability, mechanical property

## Abstract

Acute thrombosis remains the main limitation of small-diameter vascular grafts (inner diameter <6 mm) for bridging and bypassing of small arteries defects and occlusion. The use of hydrogel tubes represents a promising strategy. However, their low mechanical strength and high swelling tendency may limit their further application. In the present study, a hydrogel vascular graft of Ca alginate/polyacrylamide reinforced with a braided fiber strut was designed and fabricated with the assistance of a customized casting mold. Morphology, structure, swellability, mechanical properties, cyto- and hemocompatibility of the reinforced graft were characterized. The results showed that the reinforced graft was transparent and robust, with a smooth surface. Scanning electron microscopic examination confirmed a uniform porous structure throughout the hydrogel. The swelling of the reinforced grafts could be controlled to 100%, obtaining clinically satisfactory mechanical properties. In particular, the dynamic circumferential compliance reached (1.7 ± 0.1)%/100 mmHg for 50–90 mmHg, a value significantly higher than that of expanded polytetrafluoroethylene (ePTFE) vascular grafts. Biological tests revealed that the reinforced graft was non-cytotoxic and had a low hemolysis percentage (HP) corresponding to (0.9 ± 0.2)%. In summary, the braided fiber-reinforced hydrogel vascular grafts demonstrated both physical and biological superiority, suggesting their suitability for vascular grafts.

## 1. Introduction

Cardiovascular diseases (CVDs) are the leading cause of human mortality worldwide, and the population of CVD patients has been increasing exponentially [1,2]. Polyethylene terephthalate (PET) and expanded polytetrafluoroethylene (ePTFE) are the most commonly used materials for large-diameter vascular grafts, such as an endovascular stent and graft [3,4]. However, small-diameter vascular grafts (inner diameter <6mm), which are highly demanded coronary artery bypasses for grafting or substituting lower limb arteries, are prone to causing thrombus formation, stenosis, pseudo-endothelial hyperplasia, and consequent luminal occlusion [5,6,7]. Thus, current commercial products are still far from ideal, and thrombosis is the primary issue limiting their intermediate and long-term utility [8].

Tissue-engineered small-diameter vascular grafts (TESDVGs) have demonstrated to be promising for CVD treatment and have been extensively explored in the past few decades [9]. Among the critical elements of TESDVG (e.g., scaffolds, cells, and growth factors (GFs)) [10], fabrication of a porous scaffold is relatively straightforward using established methods such as freeze-drying, phase separation, particulate leaching, gas foaming, and so on [11,12,13]. However, it remains a significant challenge to identify the appropriate cells for regenerating new extracellular matrix close to the native one, while preventing programmed cell death or apoptosis during implantation [14]. Furthermore, the involvement of multiple GFs in the spatiotemporal regulation of cellular activities also determines the necessity of synergistically incorporating these GFs in the creation of TESDVGs [15,16]. Up to date, it is not yet imaginable to design and construct a GF-induced, orchestrated cascade system. In vitro dynamic culture could help improve the functionality of TESDVGs [7]. However, deviation from in vivo circumstances in terms of mechanical and biological environments occurs. There is still a long way to go for TESDVGs before their broad adoption for clinical applications.

Because of their hydrophilicity and biocompatibility, hydrogels have found multifaceted applications in biomedicine [17,18], for example, as scaffolding materials for tissue formation, vehicles for drug and growth factor delivery [19,20,21,22], protection layer for encapsulated cells [23,24,25], and so on [21,26,27]. In particular, in consideration of their anti-coagulation and hemocompatibility [28], hydrogels may find great use in vascular grafts. However, some associated weaknesses such as poor mechanical properties and high swelling capacities should be addressed. Thus, increasing efforts have been made to improve the mechanical properties and control the swellability of hydrogels [29,30].

An ideal small-diameter vascular graft should have specific features concerning mechanical properties (suture retention, longitudinal and circumferential compliance, tensile and compressive strength, and so forth), biological performance (cyto- and hemocompatibility), blood permeability, biodegradability, morphological and structural stability, ease for processability, sterilization, packaging, storage, and transit performance. Therefore, in the present study, sodium alginate and acrylamide (AAm) were employed to fabricate an interpenetrating double-network hydrogel with a three-step process modified by a previous method [29,31,32,33]. By combination with a PET braided fiber strut, the mechanical properties of the hydrogel tube were enhanced significantly to meet the operational requirements in terms of tensile strength, tensile elongation, circumferential compliance, compressive strength, and suture retention. Furthermore, the swellability and inner diameter of the reinforced hydrogel vascular grafts could be maintained in an aqueous environment within the test time. As expected, the reinforced hydrogel tubes exhibited high cytocompatibility and meager hemolysis percentage (HP).

## 2. Materials and Methods

### 2.1. Design and Fabrication of a 3D Casting Mold

A stainless-steel casting mold was explicitly designed and fabricated to form a braided fiber strut reinforced hydrogel vascular graft. Briefly, the mold was composed of seven parts, i.e., base, base adaptor, lower adaptor, upper adaptor, rod, and two half pipes (Figure 1). The inner and outer diameters of the reinforced graft were designed to be 6 and 12 mm, respectively. Thus, the diameter of the rod was 6 mm. The outer diameters of the lower and upper adaptors and the inner diameter of the braided fiber strut were the same, i.e., 9 mm.

### 2.2. Preparation of the Ca Alginate/Polyacrylamide Hydrogel

The Ca alginate/polyacrylamide (PAAm) hydrogels were prepared as described previously [29] with some modifications. Three steps were involved. First, acrylamide (AAm) and sodium alginate (SA) were dissolved to form an aqueous solution. Ammonium persulfate (APS) as a photo-initiator, *N, N’*-methylenebisacrylamide (MBAA) as a crosslinker, and *N, N, N’, N’*-tetramethylethylenediamine (TEMED) as an accelerator for polyacrylamide, were mixed into the aqueous solution to form a pre-gel. Second, the pre-gel was immersed into 1.0 M CaCl_2_ aqueous solution to crosslink the alginate. Third, the crosslinked hydrogel was soaked in distilled water for 12 h to remove the residual monomers and finally obtain the Ca alginate/polyacrylamide hydrogel. The concentrations used were as follows: 16.8% (w/v) for AAm, 2.3% (w/v) for SA, 0.1% (w/v) for APS, 0.01% (w/v) for MBAA, and 0.09% (w/v) for TEMED. Various immersion times (3h, 4h, 5h, and 6h) of CaCl_2_ aqueous solution were tested to identify the optimum crosslinking time, and the samples were correspondingly labeled as S1, S2, S3, and S4. All the raw materials were provided by Sinopharm Chemical Reagent Co., Ltd., Shanghai, China.

### 2.3. Scanning Electron Microscopy (SEM) of the Hydrogels

The hydrogels with a size of 30 mm × 3 mm × 3 mm were freeze-dried by liquid nitrogen and broken into half for exposure of cross sections to SEM. Gold powder was sprayed on the samples prior to microscopy. A scanning electron microscope (TM 3000, Hitachi, Tokyo, Japan) was used to examine the microscopic morphology of the hydrogels. The accelerating voltage was 15 kV.

### 2.4. Volume Swellability of the Hydrogels

Swellability can be characterized by the swelling ratio in weight and in volume [30,34]. Since the focus of the present work was to attain a vascular graft with higher mechanical properties and lower swellability to maintain a constant diameter during implantation, the swelling ratio in volume was more important. The swelling ratio (SR) in volume was calculated according to Equation (1) [30]:(1)$$\mathrm{SR}\left(\%\right)=\frac{V}{V_{0}}\;\times 100$$
where SR is the swelling ratio, V is the hydrogel volume upon swelling equilibrium, and V_0_ is the initial hydrogel volume.

For measurement, the as-prepared hydrogels were cut into strips with a size of 50 mm × 20 mm × 3 mm. Volumes could be calculated by measuring the length, the width, and the height of the bulk hydrogels.

### 2.5. Tensile Strength and Elongation of the Hydrogels

The tensile strength of the hydrogels was tested on an automatic versatile mechanical property tester (YG(B)026G-500, Darong Textile Instrument Co., Ltd., Wenzhou, China) following the Chinese National standard GB/T 528-2009. The hydrogels reaching swelling equilibrium (swelling for 24 h) were cut into strips with a size of 50 mm × 20 mm × 3 mm, the gap was set to 25 mm, and the stretching speed was set to 500 mm/min.

### 2.6. Design and Fabrication of a Braided Fiber Strut

PET monofilaments (medical grade) with a diameter of 0.25 mm (Suzhou Suture Needle Company, Suzhou, China) were braided with a 32-bobbin braiding machine to form a tube with the inner diameter of 9 mm. The braided fiber strut on a rod was then heated at 190 °C for 10 min. Varying braiding angles (50°, 55°, and 60°) were chosen to adjust the strain of the braided fiber strut to match the strain of the hydrogel tube. Deformation of the braided fiber strut and of the hydrogel tube and their hybrid were evaluated using diameter reduction rate (DR) %, which was calculated following Equation (2):(2)$$\mathrm{DR}\left(\%\right)=\frac{D_{i}-D_{d}}{D_{i}}\times 100$$
where DR (%) is the diameter reduction rate (%), D_i_ is the initial inner diameter of a tube, and D_d_ is the inner diameter of the tube when strained at 50%.

### 2.7. Fabrication of Braided Fiber Strut-Reinforced Hydrogel Vascular Grafts

With the assistance of a mold, a braided fiber strut reinforced hydrogel vascular graft was fabricated from the Ca alginate/polyacrylamide hydrogel and PET strut (Figure 2). Briefly, the braided fiber strut was wrapped around the rod, and both ends of the braided fiber strut were tied to the upper adaptor and the lower adaptor. Two half pipes were combined to form a complete pipe outside the braided fiber strut, and then the pre-gel aqueous solution was poured into the cavity between the rod and the pipe. Upon pre-gelation, the two half pipes were removed, and the pre-gel structure was soaked into 1.0 M CaCl_2_ aqueous solution for crosslinking, followed by water immersion to remove the residuals. Actually, the pre-gel was a network of PAAm embedding alginate chains. Afterward, the rod, the upper adaptor, and the lower adaptor were removed to release the braided fiber strut-reinforced hydrogel vascular grafts.

### 2.8. Mechanical Properties of the Reinforced Grafts

The mechanical properties of the reinforced graft were similarly tested for tensile strength, dynamic circumferential compliance, compressive strength, and suture retention force as previously described [35,36]. Briefly, the tensile strength was longitudinally tested on an automatic versatile mechanical property tester (YG(B)026G-500, Darong Textile Instrument Co., Ltd., Wenzhou, China) by Chinese national standard GB/T 528-2009. The maximum strain was set to 50%. The dynamic circumferential compliance was tested under a sinusoidal wave formed at a 1 aHz frequency with a BioDynamicTM Test Instrument (Bose Corporation, The Mountain Framingham, MA, USA) using the following three pressure ranges: 50–90, 80–120, and 110–150 mmHg, in accordance to the standard ISO 7198:2016. The compressive strength was circumferentially tested on a specially designed radial compression measuring apparatus (Model LLY-06D, Laizhou Electronic Instrument Co., Ltd., Laizhou, China) according to ISO 25539-2:2012. The presser foot of the apparatus was 5 mm in diameter, and the reinforced graft was compressed to 50% of the outer diameter and then released at the same rate of 10 mm/min. The displacement of the presser foot was recorded to calculate elastic recovery. The suture retention force was measured on a universal mechanical property tester (Model YG-B 026H, Laizhou Electronic Instrument Co., Ltd., Laizhou, China) following the standard ISO 7198: 2016. A 2-0 braided polyester suture (Jinhuan Medical Supplies Co. Ltd., Shanghai, China) was used. The suture was passed through the reinforced graft at the site exactly 2 mm away from the edge of the end. Then, the suture was pulled at a speed of 50 mm/min until the specimen failed. Tubular hydrogels were also prepared and characterized as control samples.

For the swellability tests, the grafts were cut into segments with a size of 50 mm in length. Volumes could be calculated by measuring the inner and outer diameters of the tubes.

### 2.9. In Vitro Cytotoxicity Assay of the Reinforced Grafts

Porcine iliac artery endothelial cells (PIECs, Cell Bank, Shanghai Institutes for Biological Sciences, Chinese Academy of Sciences, Shanghai, China) were cultured in Dulbecco’s Modified Eagle Medium (DMEM) with 10% fetal bovine serum and 1% penicillin/streptomycin (complete culture media, Thermo Fisher Scientific, Carlsbad, CA, USA) in a 37 °C, 5% CO_2_ environment. Conditioned cell culture media (CM) was prepared by incubating the reinforced grafts in the media for 24 h to determine the effect of grafts eluates on PIECs. According to the standard ISO 10993-12:2012, 1 g of the reinforced graft was put into 10 mL cell culture media in a 37 °C, 5% CO_2_ environment to prepare CM. The WST (water-soluble tetrazolium) assay was performed with CCK-8 (Yeasen Biotechnology Co., Ltd., Shanghai, China) according to the manufacturer’s instructions. In total, 2000 cells/well were seeded in 100 μL complete culture media in 96-well plates for 4 h before changing to 100 μL CM. The absorbance was measured at 450 nm using a BioTek plate reader (BioTek, Winooski, VT, USA). Relative proliferation rate (RPR, %) was calculated on the basis of optical density values by equation (3):(3)$$\mathrm{RPR}\left(\%\right)=\frac{OD_{s}}{OD_{c}}\times 100$$
where OD_s_ is the absorbance of the sample, and OD_c_ is the absorbance of the negative controls.

### 2.10. Hemolysis Test of the Graft

Fresh whole blood obtained from New Zealand white rabbits’ ear veins was anticoagulated and utilized to test the hemolytic performance of the reinforced grafts. The blood was centrifuged (Biofuge Primo Model R centrifuge, Thermo Fisher Scientific, Carlsbad, CA, USA) and washed with PBS five times following standard procedures as reported [37,38]. A volume of 1 mL of red blood cells (RBCs) was suspended in 34 mL PBS. Then, 0.2 mL of the RBCs suspension was transferred to a 5 mL Eppendorf tube, which was filled with either 0.8 mL of deionized water as the positive control, or PBS buffer as the negative control. The reinforced graft samples were incubated in the suspension containing 0.2 mL of diluted RBCs and 0.8 mL of PBS buffer at 37 °C for 2 h, followed by centrifugation for 3 min at 10,000 rpm with an Eppendorf 5415 Model R centrifuge (Eppendorf, Hamburg, Germany). Then, the optical density of the supernatant was determined by a Perkin Elmer Lambda 25 UV–visible spectrophotometer (Perkin Elmer, Waltham, MA, USA) at 540 nm. The HP was calculated using Equation (4) [39,40]:(4)$$\mathrm{HP}\left(\%\right)=\frac{D_{s}-D_{nc}}{D_{pc}-D_{nc}}\times 100$$
where D_s_ is the absorbance of the sample, and D_pc_ and D_nc_ are the absorbances of the positive and negative controls, respectively.

### 2.11. Statistical Analysis

All data obtained in the present work were expressed as mean ± standard deviation (SD), and a significant difference was analyzed using ANOVA and unpaired student t-test; *p* < 0.05 was considered significant.

## 3. Results and Discussion

### 3.1. Morphology and Porous Structure of the Ca Alginate/Polyacrylamide Hydrogel

The Ca alginate/polyacrylamide hydrogel prepared in the present work was transparent and smooth (Figure 3a). The hydrogel was stretchable, flexible and elastic when tested for tensile and compressive strength. SEM examination revealed a porous structure of the hydrogel (Figure 3b). A wide range of pore size (pore area), i.e., 287–6150 μm^2^ was measured, and the average pore size was 3556 ± 995 μm^2^ (Figure 3c). In the present work, the as-prepared pre-gel was soaked in a CaCl_2_ aqueous solution for gelation instead of adding the CaCl_2_ solution into the mixed solution [41] or soaking in the CaSO_4_·2H_2_O aqueous solution [29], so the gelation rate could be tuned, which led to a homogeneous hydrogel.

### 3.2. Tensile Strength of the Ca Alginate/Polyacrylamide Hydrogel

A series of tensile tests were performed on the hydrogels with different immersion times (S1: 3h, S2: 4h, S3: 5h, and S4: 6h) to determine the optimal crosslinking time. Figure 4 shows the results of the tensile breaking strength and tensile breaking elongation of four types of hydrogel (S1–S4) (n = 3). The tensile breaking strengths of S1 and S2 decreased significantly after reaching the swelling equilibrium compared to those of the as-prepared hydrogels (Figure 4a). Meanwhile, a significant increase in water content and volume were seen for S1 and S2 hydrogels upon reaching the swelling equilibrium. While no apparent changes were observed for S3 and S4 before and after swelling equilibrium, the tensile breaking strength of S3 was much higher than those of S1 and S2. This phenomenon could be explained by the fact that the ionic crosslinking increased over prolonged immersion times till 5 h, and then the crosslinking of G blocks of alginate with Ca^2+^ reached its plateau by 5 h; as a result, the tensile strength increased with the increase of the crosslinking degree.

Figure 4b shows that the tensile breaking elongations of the four hydrogels decreased significantly because of swelling compared to those of the as-prepared hydrogels. However, no significant difference was identified among four hydrogels either before or after swelling, suggesting that the immersion time in CaCl_2_ solution had a minimal effect on the elongation performance of the hydrogels. On the one hand, the amount of Ca^2+^ was meager for crosslinking; on the other hand, the influence of the difference of immersion time on crosslinking saturation could not be shown, since 3 h might be enough for almost complete crosslinking. Combined with the above results, the optimal immersion time selected was 5 h.

### 3.3. Volume Swellability of the Ca Alginate/Polyacrylamide Hydrogel

Representative swelling properties of the hydrogels as a function of time in deionized water are presented in Figure 5a. The results showed that the swelling ratios of all four types of hydrogels increased as a function of the immersion time (3 to 6 h) in deionized water until equilibrium. Meanwhile, the equilibrium times for hydrogels S1, S2, S3, and S4 were determined as 12, 12, 0, and 0 h, respectively. In other words, no volume changes were found for S3 and S4; that is, the swelling ratios were 100%. Figure 5b showed that the swelling ratios of S1 and S2 were significantly higher than those of S3 and S4 when they reached the swelling equilibrium. This observation indicates that vascular grafts derived from both S3 and S4 would have satisfying structural stability, and the volume and configuration (e.g., the inner diameter) would remain constant. This point is of critical importance for hydrogel-based vascular grafts, especially for small-diameter vascular grafts, as any subtle change to the diameter of vascular grafts may evoke enormous turbulence of the blood flow, which would induce acute thrombosis and, consequently, luminal occlusion [41]. These results are in good agreement with those of the mechanical properties. In combination, S3 would be an excellent choice to fabricate small-diameter vascular grafts possessing not only robust mechanical properties but also constant structural stability, suitable for in vivo implantation. Hence, S3 was selected for the following experiments.

### 3.4. Fabrication of Strut-Reinforced Hydrogel Vascular Grafts

With the assistance of a customized stainless-steel casting mold (Figure 6a), hydrogel-based vascular grafts (Figure 6d) reinforced with a braided fiber strut (Figure 6b) could be fabricated. To maintain the integrity between hydrogel and fiber strut, it became necessary to assure a similar deformation behavior of hydrogel tubes and fiber struts. In this regard, several fiber struts with different braiding angles (Table 1) were chosen and compared to S3 hydrogel tubes. In consideration of the typical shrinking rate of a native blood vessel of about 33% [42], an elongation rate of 50% was used for the tensile test. As shown in Table 1, the braided fiber strut with a braiding angle 55° had a deformation comparable to that of the S3 hydrogel tube (Table 1, n = 3). Therefore, this braided fiber strut was selected to fabricate reinforced hydrogel vascular grafts for the following experiments.

With the desired hydrogel and braided fiber struts, strut-reinforced hydrogel vascular grafts could be readily fabricated. During the fabrication, the inner diameter of the vascular graft and the thickness of the wall could be precisely adjusted and controlled by adjusting the mold parameters. Moreover, the braided fiber strut could be positioned in the middle, inner, or outer layer of the wall. Compared to other crimp small-diameter vascular grafts made of PET and ePTFE, a prominent advantage of the vascular graft described in the present work is that the lumen would not close while bending even without a crimp wall, because its morphology is similar to that of a native blood vessel possessing a layered thick and elastic wall. Hydrogel tubes without a braided fiber strut (Figure 6c) were also fabricated as controls.

### 3.5. Mechanical Properties of the Strut-Reinforced Hydrogel Grafts

Native blood vessels are strained inside the body and they are shortened upon cut-off from the primary site, e.g., the length of a swine carotid artery is reduced to 2/3 of its original length upon cut-off from the body [42]. Therefore, the length of an implanted vascular graft should refer to the original length of a target native blood vessel, and, ideally, the graft should have the same elongation rate as the native vessels. The tensile properties of the reinforced grafts were evaluated, setting the strain to 50%. The results showed that the incorporation of a braided fiber strut significantly improved the mechanical properties of the grafts (Figure 7), and the reinforced graft could almost completely recover to its original shape upon relief of the force. More importantly, a simultaneous response of the braided fiber strut and the hydrogel tube was invariably found, without any mismatch or interfacial separations. The measured stress of the reinforced grafts was almost the sum of those of the braided fiber strut and of the hydrogel tube and reached 201.3 ± 6.91 kPa with a set strain of 50%, close to but still higher than that of a swine carotid artery (*p* < 0.05) (Figure 7 inset). This observation implies that the strut-reinforced grafts can meet the tensile strength requirements for clinical application.

The compliance of a vascular graft plays a pivotal role in maintaining a healthy blood flow, avoiding abnormal tissue hyperplasia and thrombosis [43,44]. However, the compliance of the existing vascular grafts made of either PET or ePTFE was low, corresponding to less than 1% [41], which partly accounts for the unsuitability of such grafts for small-diameter vessels [45]. The circumferential compliances of the reinforced graft, hydrogel tube, and a commercial ePTFE vascular graft were respectively tested according to the standard ISO 7198: 2016 on the BOSE platform under three spans of pressures, i.e., 50–90, 80–120, and 110–150 mmHg. The measurement revealed that the compliances of the hydrogel tubes and the reinforced grafts were much higher than that of the ePTFE vascular grafts (*p* < 0.05) in all three pressure conditions, despite a decreased compliance of the reinforced grafts by the braided fiber strut (Figure 8). The compliance of reinforced hydrogel grafts under a pressure of 50–90 mmHg reached (1.7 ± 0.1)%/100 mmHg, significantly higher than that of the ePTFE vascular grafts, indicating the superiority of reinforced hydrogel grafts for the substitution or bypass of small-diameter blood vessels.

In addition to the tensile strength, the compressive strength of both the hydrogel tube and the reinforced graft was tested. As shown in Figure 9, the incorporation of the braided fiber strut in the reinforced grafts significantly increased the compressive strength of the grafts, demonstrating the vital role of the braided fiber strut in strengthening the radial mechanical properties. Upon relief after compression to 50% of the outer diameter, the radial elastic recovery could reach (90.1 ± 1.3)%.

Considering suturing is the first step of surgery, suture retention force is another essential parameter for the mechanical characterization of vascular grafts. Thus, a vascular graft should be able to suture to the native blood vessels. The suture retention force measurement showed that the reinforced graft yielded 8.4 ± 0.5 N retention force, remarkably higher than those of a canine femur artery (7.9 ± 0.4 N, *p* < 0.05) [46] or of hydrogel tubes (6.8 ± 1.3 N, *p* < 0.05). This finding implies that the braided fiber strut would be the dominant contributor to strengthen the suture retention force. Taken together, it becomes clear that the reinforced grafts can not only meet the mechanical need for surgery but also provide satisfactory structural flexibility.

### 3.6. Swellability of the Strut-Reinforced Vascular Grafts

The swellability of grafts, especially upon implantation, is critical to graft stability [47]. Thus, the swellability and inner diameter of hydrogel (S3) tubes and braided fiber strut-reinforced vascular grafts were evaluated for 24 h. As shown in Figure 10, the swellability and inner diameters of both samples kept constant. The swellability remained 100% for the entire testing period, with no volume changes. The inner diameters also remained regularly at 6.5 mm. These observations indicate that the strut-reinforced vascular grafts would maintain their structural stability during implantation.

### 3.7. In Vitro Cellular Viability of the Strut-Reinforced Vascular Grafts

Strut-reinforced vascular grafts sterilized by autoclave were incubated in cell culture media for 24 h to prepare the conditioned cell culture media (CM). PIECs were divided into two groups, i.e., graft-incubated group (CM) and control group (CG, complete culture media). The cells cultured in either CM or CG media for 1, 3, and 5 days were analyzed with the WST assay. As shown in Figure 11, the optical density (OD), proportional to the cell number, increased over the culture time, and the relative proliferation ratio (RPR, %) in CM and CG for 1, 3, and 5 days was 81.3%, 90.6%, and 93.7%, respectively. On the basis of the criteria for cytocompatibility [48], the strut-reinforced vascular graft can be defined as non-cytotoxicity, consistently with previously reported results [31].

### 3.8. Hemolysis Percentage of the Reinforced Graft

Vascular grafts are also required to be hemocompatible. HP is usually used to evaluate the destructive properties of biomaterials with respect to erythrocytes, especially for blood contact biomaterials [49]. The results of the hemolysis tests for the strut-reinforced graft showed that its HP was as low as (0.9 ± 0.2)%. Thus, it can be considered as a non-hemolytic material according to the standard ASTM F756-00, where the non-hemolytic range is defined as 0–2% [50,51]. On the basis of both biological test results, the strut-reinforced vascular graft showed satisfactory cyto- and hemocompatibilities. In summary, the newly developed vascular grafts have not only robust mechanical properties but also stable structure and excellent biocompatibility. In particular, they have a much higher circumferential compliance and lower swellability in volume.

## 4. Conclusions

A braided PET tubular strut-reinforced Ca alginate/polyacrylamide hydrogel vascular graft was fabricated with the assistance of a customized 3D casting mold. The mechanical properties concerning tensile strength, tensile elongation, dynamic circumferential compliance, compressive strength, and suture retention performance of the strut-reinforced grafts were improved significantly in comparison to native blood vessels and ePTFE prostheses. Furthermore, the shape of strut-reinforced grafts could be maintained upon prolonged incubation in an aqueous environment. Thus, this strut-reinforced graft holds great promise to meet the necessary requisites for clinical operations. More importantly, it has a satisfying cytocompatibility, hemocompatibility, suturing retention, and dynamic circumferential compliance. The present work paves the way to develop a new product-oriented small-diameter vascular graft.

## Figures and Tables

**Figure 1 polymers-11-00810-f001:**
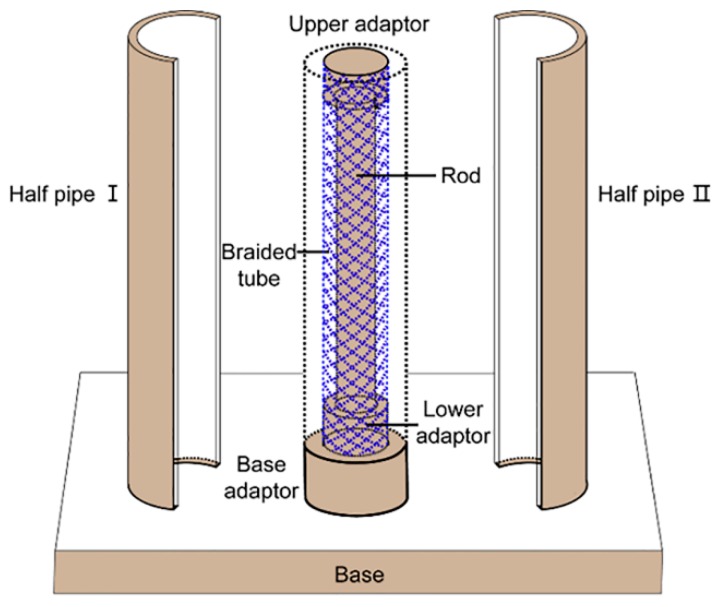
Schematic diagram of the stereo-casting mold.

**Figure 2 polymers-11-00810-f002:**
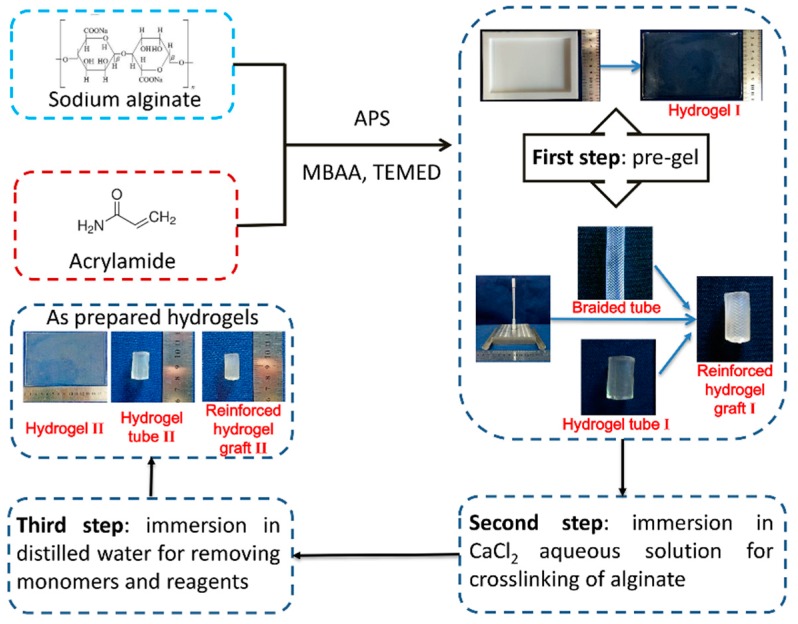
Schematic illustration of the preparation process of the reinforced hydrogel vascular graft. APS: ammonium persulfate, MBAA: N, *N’*-methylenebisacrylamide, TEMED: *N, N, N’, N’*-tetramethylethylenediamine.

**Figure 3 polymers-11-00810-f003:**
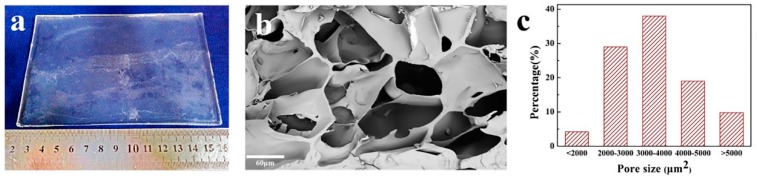
Images of the Ca alginate/polyacrylamide hydrogels (**a**), SEM microphotograph of a cross section (**b**) and distribution of the pore sizes (**c**).

**Figure 4 polymers-11-00810-f004:**
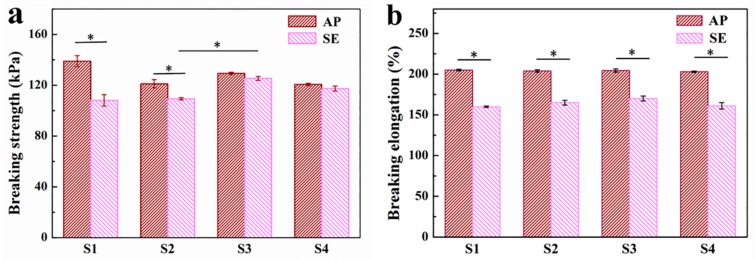
Results of the tensile breaking strength and tensile breaking elongation tests of four hydrogels (S1: 3h; S2: 4h; S3: 5h; S4: 6h). (**a**) Tensile breaking strength. (**b**) Tensile breaking elongation; * means significant difference at *p* < 0.05, AP: as-prepared hydrogels, SE: hydrogels at swelling equilibrium.

**Figure 5 polymers-11-00810-f005:**
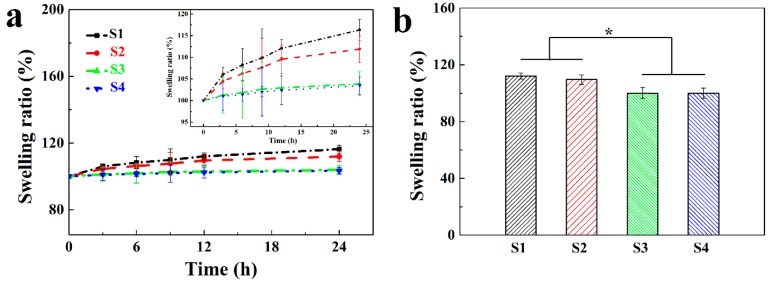
Swellability of four hydrogels. Processes of approaching the swelling equilibrium (**a**), equilibrium swelling ratios (**b**) (* *p* < 0. 05).

**Figure 6 polymers-11-00810-f006:**
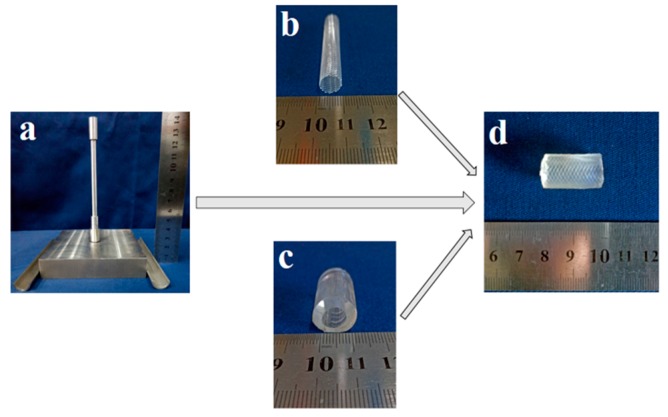
Images of the stereo-casting mold (**a**), braided polyethylene terephthalate (PET) tube (**b**), hydrogel tube (**c**), and reinforced hydrogel graft (**d**).

**Figure 7 polymers-11-00810-f007:**
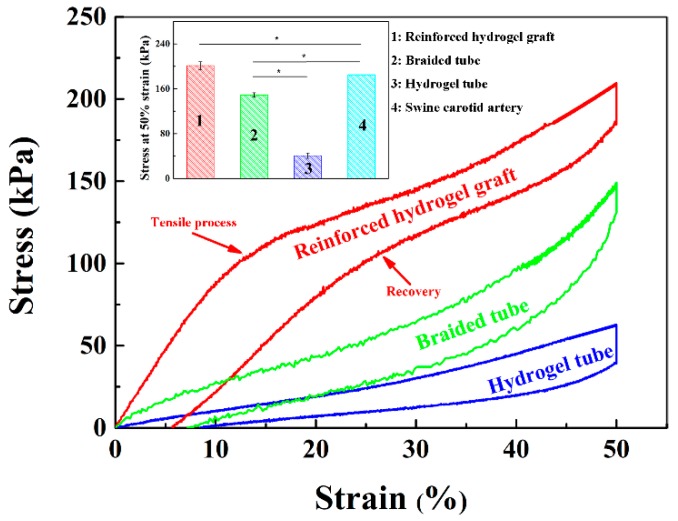
Strain-–tress tensile curves of the strut-reinforced hydrogel grafts, hydrogel tubes, and braided fiber struts. Inset: results of the quantitative analysis.

**Figure 8 polymers-11-00810-f008:**
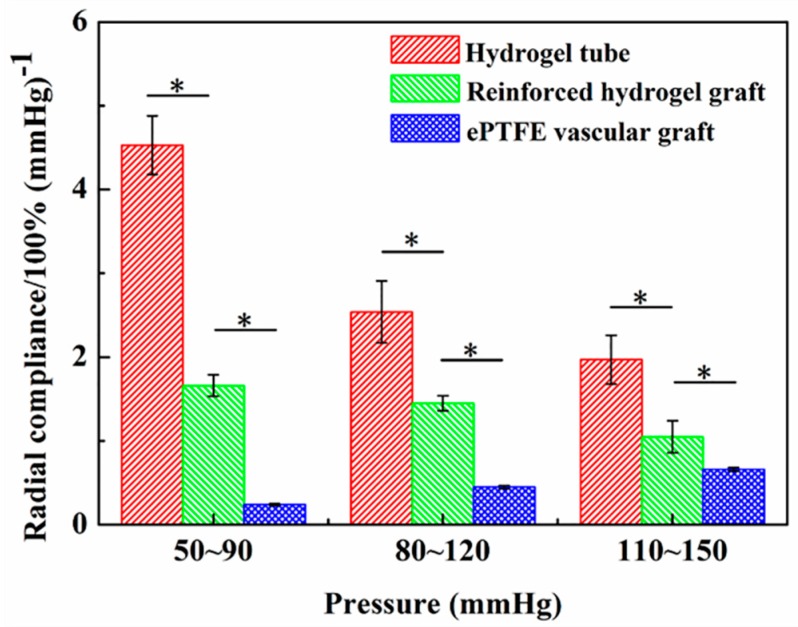
Circumferential compliance of the hydrogel tubes, strut-reinforced hydrogel grafts, and an approved expanded polytetrafluoroethylene (ePTFE) vascular graft (n=3).

**Figure 9 polymers-11-00810-f009:**
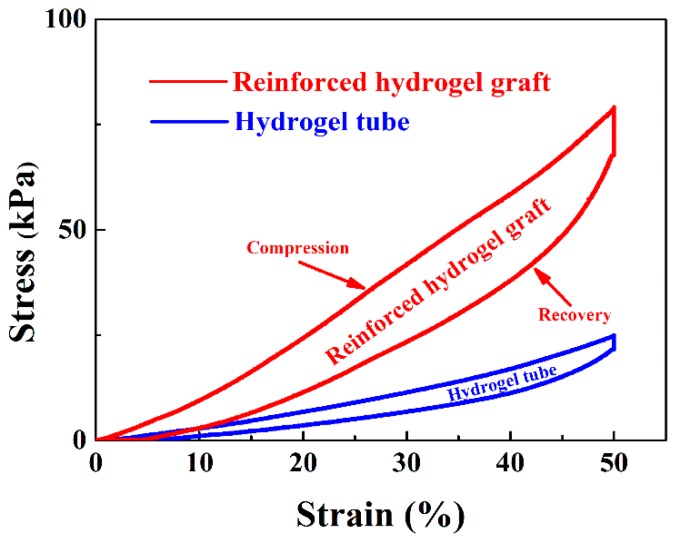
Compressive strength of the strut-reinforced grafts and hydrogel tubes.

**Figure 10 polymers-11-00810-f010:**
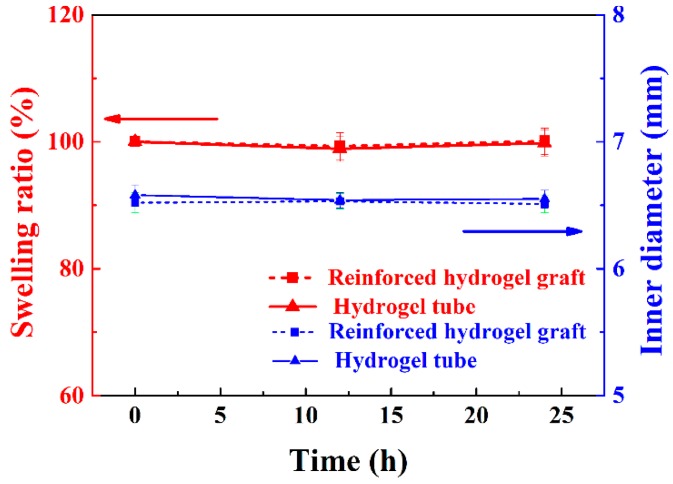
Evolution of swellability in volume and inner diameters of the reinforced hydrogel grafts and hydrogel tubes over time.

**Figure 11 polymers-11-00810-f011:**
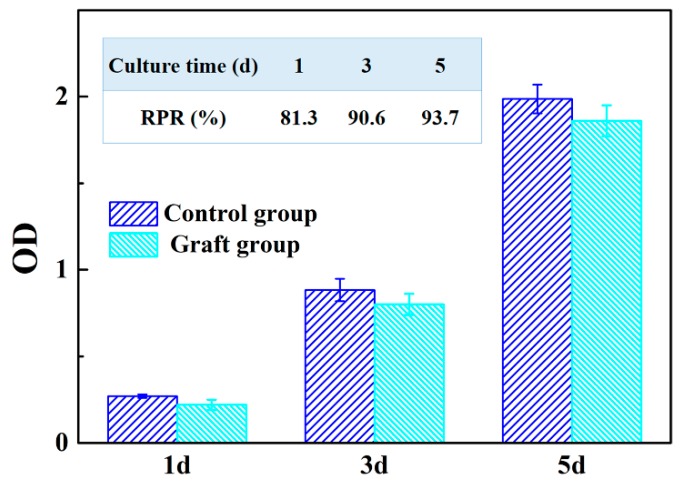
Results of cell culture in conditioned cell culture media (CM) and complete culture media. RPR: relative proliferation ratio.

**Table 1 polymers-11-00810-t001:** Diameter reduction rate (%) of the braided fiber struts and S3 hydrogel tubes.

Number	Braiding Angle (°)	Diameter Reduction Rate (%)
1	50 ± 0.5	26.52 ± 0.64
2	55 ± 0.5	18.73 ± 0.68
3	60 ± 0.5	7.98 ± 0.22
S3	/	18.14 ± 0.94

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
