# Peer review of "Hydrogel Small-Diameter Vascular Graft Reinforced with a Braided Fiber Strut with Improved Mechanical Properties"

_polymers, 2019, doi:10.3390/polym11050810_

Round 1

Reviewer 1 Report

The work by Guan et al. reports on the design of new small diameter vascular grafts based on a hydrogel reinforced with a braided fiber structure. In detail, the two components of the device, i.e. an alginate/polyacrylamide hydrogel and a PET-based fibrous structure, were first designed and characterized and then the final composite system was assembled by using a custom-made 3D casting mold allowing the fabrication of tubular structures with diameter of approx. 6 mm in which the reinforcing fiber structure was embedded in the hydrogel. The paper reports an innovative approach to design small diameter vascular grafts with proper mechanical and swellability properties for the intended application. Results have been clearly reported. Moreover, the conclusions were adequately supported by both the obtained results and literature reported data. English language and style are fine and minor spell check is required to completely eliminate errors (e.g., line 27: “demonstrated” instead of demonstrate, line 99 reformulate “during implanted”, line 216 correct “that that”). Although the high quality of the work, some major revisions are still required before acceptance for publication in Polymers. A list of the suggested revisions will follows.

1. the abstract should briefly report on the preparation of the hydrogel, the fibrous structure and the assembly of the whole device. Additionally, the main results should be reported also in the abstract to clearly state the quality of the obtained data

2. the introduction should thoroughly report the requirements a small diameter vascular graft should possess

3. the hydrogels designed in this work have not been “synthesized” as they result from the blending and cross-linking of commercially available monomers/polymers. Instead of “synthesis”, the authors should speak about “preparation”

4. Why a photo-initiator has been added to the precursors solution? ammonium persulfate is usually used as catalyst, while TEMED is used as initiation of the radical polymerization leading to polyacrylamide

5.  line 18: change “polyacrylamide” with “AAm crosslinking”

6. the authors should clarify the nature of the pre-gel. It seems to be a network of polyacrylamide embedding alginate chains. Is this right?

7. a whole reorganization of “materials and methods” section should be done to make reading easier and adequately present the flowchart of the work. The reviewer suggests this organization:

3D casting mold à Preparation of Ca-alginate/polyacrylamide hydrogels à characterization of hydrogels (SEM, swellability, mechanical testing) à design of the fiber structures à their characterization (mechanical test) à fabrication of the final devices à characterization of the vascular grafts (as reported in paragraphs 2.8, 2.9 and 2.10) à  statistical analysis

8. details on sample preparation and analysis conditions during SEM should be added

9. references to the characterization of the tubular systems should be moved where required and not embedded in the paragraphs dedicated to hydrogel bulk characterization. The authors should state that tubular hydrogels have been also prepared and characterized as control samples

10. the equation required to calculate RPR% should be added in paragraph 2.9

11. swellability tests on the final devices should be described in paragraph 2.8

12. the term “pore size” usually refers to pore diameter. The authors should clearly state they are talking about pore area

13. SEM should be performed on S1, S2, S3 and S4, on as prepared and equilibrium swollen hydrogels

14. the authors should report in materials and methods that mechanical testing was performed on as prepared and equilibrium swollen gels (please report also the time of swelling adopted prior to characterization)

15. Did the authors evaluate the effects of sterilization on the devices?

16. Swellability of the final devices should be tested for longer time intervals than 24h

17. in the conclusions the authors should further stress that the newly devices work very well upon suture

Author Response

Dear Reviewer,

Thank you very much for your kind consideration to review our work. We have carefully revised our manuscript based on your comments, and marked details point-to-point in response to the comments. We hope that the revised manuscript would be acceptable for the publication in Polymers. Again the authors appreciate your time and efforts in handling our manuscript.

Manuscript ID: polymers-493648

Title: Hydrogel small diameter vascular graft reinforced with a braided fiber strut for improved mechanical properties

The work by Guan et al. reports on the design of new small diameter vascular grafts based on a hydrogel reinforced with a braided fiber structure. In detail, the two components of the device, i.e. an alginate/polyacrylamide hydrogel and a PET-based fibrous structure, were first designed and characterized and then the final composite system was assembled by using a custom-made 3D casting mold allowing the fabrication of tubular structures with diameter of approx. 6 mm in which the reinforcing fiber structure was embedded in the hydrogel. The paper reports an innovative approach to design small diameter vascular grafts with proper mechanical and swellability properties for the intended application. Results have been clearly reported. Moreover, the conclusions were adequately supported by both the obtained results and literature reported data. English language and style are fine and minor spell check is required to completely eliminate errors (e.g., line 27: “demonstrated” instead of demonstrate, line 99 reformulate “during implanted”, line 216 correct “that that”). Although the high quality of the work, some major revisions are still required before acceptance for publication in Polymers. A list of the suggested revisions will follow.

Answer: The authors really appreciate the reviewer’s positive response to our work. We checked through the full paper again for eliminating errors such as those mentioned above. Please see changes in the manuscript marked in red with the editing/tracking mode.

1. The abstract should briefly report on the preparation of the hydrogel, the fibrous structure and the assembly of the whole device. Additionally, the main results should be reported also in the abstract to clearly state the quality of the obtained data.

Answer: The abstract was rewritten in part by the authors to add statements that the reviewer mentioned above. Please see changes in the manuscript marked in red with the editing/tracking mode.

2. The introduction should thoroughly report the requirements a small diameter vascular graft should possess.

Answer: The authors added several sentences to report requirements a small diameter vascular graft should possess. Please see changes in the manuscript marked in red with the editing/tracking mode.

3.  The hydrogels designed in this work have not been “synthesized” as they result from the blending and cross-linking of commercially available monomers/polymers. Instead of “synthesis”, the authors should speak about “preparation”

Answer: The authors replaced all “synthesis” with “preparation”.

4.  Why a photo-initiator has been added to the precursors solution? ammonium persulfate is usually used as catalyst, while TEMED is used as initiation of the radical polymerization leading to polyacrylamide.

Answer: The method employed in the present study is a routine one and was cited from literatures. Normally, ammonium persulfate was used as initiator, MBAA was added to the precursors solution as crosslinker, and TEMED was used as catalyst (Darnell M C , Sun J Y , Mehta M , et al. Performance and biocompatibility of extremely tough alginate/polyacrylamide hydrogels[J]. Biomaterials, 2013, 34(33):8042-8048.).

5.  line 18: change “polyacrylamide” with “AAm crosslinking”.

Answer: The authors did not find “polyacrylamide” in line 18, but in line 19. Please check if we need to correct it in line 19. Thank you.

6. The authors should clarify the nature of the pre-gel. It seems to be a network of polyacrylamide embedding alginate chains. Is this right?

Answer: Yes, this is right. The authors explained the pre-gel in the same paragraph.

7.  A whole reorganization of “materials and methods” section should be done to make reading easier and adequately present the flowchart of the work. The reviewer suggests this organization:

3D casting mold à Preparation of Ca-alginate/polyacrylamide hydrogels à characterization of hydrogels (SEM, swellability, mechanical testing) à design of the fiber structures à their characterization (mechanical test) à fabrication of the final devices à characterization of the vascular grafts (as reported in paragraphs 2.8, 2.9 and 2.10) à  statistical analysis

Answer: The authors reorganized the structure according to the reviewer’s suggestions. Please find the changes in the manuscript.

8.   Details on sample preparation and analysis conditions during SEM should be added.

Answer: The authors added details on sample preparation and analysis conditions during SEM. Please find the changes in the manuscript.

9. References to the characterization of the tubular systems should be moved where required and not embedded in the paragraphs dedicated to hydrogel bulk characterization. The authors should state that tubular hydrogels have been also prepared and characterized as control samples.

Answer: The authors moved the related statements to the tubular characterization part, and stated that tubular hydrogels have been also prepared and characterized as control samples after description of the reinforced grafts.

10.  The equation required to calculate RPR% should be added in paragraph 2.9.

Answer: The equation was added in paragraph 2.9.

11.  Swellability tests on the final devices should be described in paragraph 2.8.

Answer: The statements on swellability tests on the final devices were added in paragraph 2.8.

12.  The term “pore size” usually refers to pore diameter. The authors should clearly state they are talking about pore area.

Answer: The authors added remarks for the “pore size” as it appeared for the first time, showing “pore area”.

13. SEM should be performed on S1, S2, S3 and S4, on as prepared and equilibrium swollen hydrogels.

Answer: All of S1, S2, S3 and S4, including as prepared and equilibrium swollen hydrogels were observed by SEM. Howeverthese microphotographs did not show distinct differences from one another. So the authors just selected the representative images to indicate the similar morphology, in order to avoid the redundant stacking and control the manuscript length.

14. The authors should report in materials and methods that mechanical testing was performed on as prepared and equilibrium swollen gels (please report also the time of swelling adopted prior to characterization)

Answer: The authors complemented the information in “Materials and Methods”. Please find the changes in the manuscript.

15.  Did the authors evaluate the effects of sterilization on the devices?

Answer: Yes, the authors evaluated the effects of sterilization on the devices and no significant differences can be found in terms of morphology and mechanical property. So the authors used the autoclave to sterilize the devices before cell culture.

16.  Swellability of the final devices should be tested for longer time intervals than 24h.

Answer: Indeed, the longer time intervals can be found in many previously published reports during swellability tests of hydrogels in weight or in volume, to find the turning point of the swelling or the swelling equilibrium time point. However, in the present work we found the fact that the swellability of the as prepared and the equilibrium swollen hydrogels maintained constant from 0 h through 72 h during swelling. Therefore, the authors showed 24 h results here for proving the stability of the grafts.

17.  In the conclusions the authors should further stress that the newly devices work very well upon suture.

Answer: The authors added the related information in the conclusions for addressing the outstanding suturing and compliance performance.

The authors thank the editor and both reviewers again for their diligent and careful work in reviewing the manuscript.

Sincerely,

Lu Wang, Ph.D.

Professor

Key Laboratory of Textile Science and Technology of Ministry of Education

College of Textiles, Donghua University, Shanghai, China

Reviewer 2 Report

Manuscript ID: polymers-493648-peer-review-v1

Title: Hydrogel small diameter vascular graft reinforced with a braided fiber strut for improved mechanical properties

Topic and Findings:

The authors used sodium alginate and acrylamide (AAm) to fabricate an interpenetrating double-network hydrogel to be used as vascular graft. Reinforcement was achieved by a braided fiber strut fabricated via casting mold. Morphology, structure, swellability, mechanical properties, cyto- and hemocompatibility of the reinforced grafts were characterized.

Results showed transparent and robust grafts with a smooth surface, uniform porous hydrogel structure. Swelling could be controlled; grafts are non-cytotoxic with a low hemolysis percentage.

Abstract:

Lines 15 ff.: “Acute thrombosis remains to be the main limitation to small diameter vascular grafts (inner diameter < 6 mm) for bridging and bypassing of small arteries defects and occlusion. The use of hydrogel tubes represents a promising strategy.”

Please provide a reference for the statement (regarding the inner diameter limitation to less than 6 mm).

Introduction:

Lines 63-71: “In the present study, sodium alginate and acrylamide (AAm) were employed to fabricate an interpenetrating double-network hydrogel with a three-step process modified by a previous method [29,31-33]. Combined with a PET (polyethylene terephthalate) braided fiber strut, mechanical properties of the hydrogel tube were enhanced significantly to meet the operational requirements in terms of tensile strength, tensile elongation, circumferential compliance, compressive strength, and suture retention. Furthermore, the swellability and inner diameter of the reinforced hydrogel vascular grafts could be maintained in an aqueous environment within the test time. As expected, the reinforced hydrogel tubes exhibited high cytocompatibility and meager hemolysis percentage (HP).”

Please specify more in detail, what exactly is “modified” in difference to the reported (and already published) procedure (references cited: 29,31-33). Specify more in detail your contribution and the novelty of your study.

Materials and Methods:

Lines 73 ff.: 2.1. Preparation of the Ca-alginate/polyacrylamide hydrogel

“The Ca-alginate/polyacrylamide hydrogels were synthesized as described previously [29] with some modifications.”

Please specify the supplier for the raw materials used and explain in detail, what exactly is “modified” in difference to the cited procedure (ref. 29, 31-33) – in order to proof the novelty of the current study.

e.g. in difference to_ Li, G.; Zhang, G.P.; Sun, R.; Wong, C.P. Mechanical strengthened alginate/polyacrylamide hydrogel crosslinked by barium and ferric dual ions. J. Mater. Sci. 2017, 52, 8538-8545, doi:10.1007/s10853-017-1066-x

Results and Discussion:

Line 89: 2.1. Bone Demineralization and Mechanical Tests

See above: In your introduction as state that “an optimized procedure has been developed” – so please refer to the existing procedure. Specify the novelty of the current study.

Conclusion:

Lines 359-367: “A braided PET tubular strut-reinforced Ca-alginate/polyacrylamide hydrogel vascular graft was fabricated with the assistance of a customized 3D casting mold. The mechanical properties concerning tensile strength, tensile elongation, dynamic circumferential compliance, compressive strength, and suture retention performance of the strut-reinforced grafts were improved significantly in comparison to native blood vessels and ePTFE prosthesis. Furthermore, the shape of strut-reinforced grafts could be maintained upon prolonged incubation in an aqueous environment. Thus, the strut-reinforced graft holds a great promise in meeting the required requisites for clinical operations. More importantly, it has a satisfying cytocompatibility and hemocompatibility. The resent work paves the way to develop a new product-oriented small-diameter vascular graft.”

Please provide reference data to compare the cytocompatibility data of your grafts with comparable materials (also based on Ca-alginate/polyacrylamide) reported in literature – in order to support the statement of “satisfying cytocompatibility”.

Author Response

Dear Reviewer,

Thank you very much for your kind consideration to review our work. We have carefully revised our manuscript based on your comments, and marked details point-to-point in response to the comments. We hope that the revised manuscript would be acceptable for the publication in Polymers. Again the authors appreciate your time and efforts in handling our manuscript.

Manuscript ID: polymers-493648

Title: Hydrogel small diameter vascular graft reinforced with a braided fiber strut for improved mechanical properties

Topic and Findings:

The authors used sodium alginate and acrylamide (AAm) to fabricate an interpenetrating double-network hydrogel to be used as vascular graft. Reinforcement was achieved by a braided fiber strut fabricated via casting mold. Morphology, structure, swellability, mechanical properties, cyto- and hemocompatibility of the reinforced grafts were characterized. Results showed transparent and robust grafts with a smooth surface, uniform porous hydrogel structure. Swelling could be controlled; grafts are non-cytotoxic with a low hemolysis percentage.

Answer: The authors feel grateful to the reviewer’s diligent and careful work and positive response to our work.

Abstract:

Lines 15 ff.: “Acute thrombosis remains to be the main limitation to small diameter vascular grafts (inner diameter < 6 mm) for bridging and bypassing of small arteries defects and occlusion. The use of hydrogel tubes represents a promising strategy.”

Please provide a reference for the statement (regarding the inner diameter limitation to less than 6 mm).

Answer: This classification has already been accepted widely in the field that vascular grafts with inner diameters no more than 6 mm belong to small diameter vascular grafts. Please find the reference “WANG Y Y, CHEN S Y, PAN Y W, et al. Rapid in situ endothelialization of a small diameter vascular graft with catalytic nitric oxide generation and promoted endothelial cell adhesion [J]. J. Mater. Chem. B, 2015, 3(47):9212-9222.”

Introduction:

Lines 63-71: “In the present study, sodium alginate and acrylamide (AAm) were employed to fabricate an interpenetrating double-network hydrogel with a three-step process modified by a previous method [29,31-33]. Combined with a PET (polyethylene terephthalate) braided fiber strut, mechanical properties of the hydrogel tube were enhanced significantly to meet the operational requirements in terms of tensile strength, tensile elongation, circumferential compliance, compressive strength, and suture retention. Furthermore, the swellability and inner diameter of the reinforced hydrogel vascular grafts could be maintained in an aqueous environment within the test time. As expected, the reinforced hydrogel tubes exhibited high cytocompatibility and meager hemolysis percentage (HP).”

Please specify more in detail, what exactly is “modified” in difference to the reported (and already published) procedure (references cited: 29,31-33). Specify more in detail your contribution and the novelty of your study.

Answer: The method employed in the present work referred to those reported in literatures [29,31-33], but was modified for the specific goal of preparing the tubular hydrogel grafts.

For bulk hydrogel preparation, the method reported in literatures has a procedure as follows. According to a formula, those solutions of SA, AAm, APS, MBAA, and TEMED were added into a beaker in order while stirring to ensure sufficient blending, and then CaCl2 aqueous solution was dropped into the mixed solution to crosslink alginate with vigorous stirring for uniform gelation, because the crosslinking reaction is extremely swift. Otherwise, uneven gelation would be obtained and the process must be repeated until success.

Compared to the preparation of bulk hydrogels, the difficulty for preparing tubular hydrogel grafts increased sharply, because it is difficult to add vigorous stirring on the process. Therefore, the authors tried varying ways to control the crosslinking reaction rate, such as low concentration of CaCl2 aqueous solution, BaSO4 aqueous solution, reducing the dropping speed and so forth. However, all these ways did not work well for the purpose.

Finally, the authors changed the way for crosslinking alginate, used immersion method to replace the dropping method. Moreover, elongated crosslinking time was allowed in the process, from 3h to 6 h. This modification to the method reported in literatures obtained ideal results that we obtained homogeneous hydrogel, transparent and smooth.

In the process of preparing tubular hydrogel grafts, this procedure includes three steps. Briefly, the first step is pre-gelation, the second is immersion for crosslinking alginate, and the third is removal of residual unreacted molecules or chemicals.

Detailed protocols were described briefly in the part “Materials and Methods”. Please see the manuscript.

Materials and Methods:

Lines 73 ff.: 2.1. Preparation of the Ca-alginate/polyacrylamide hydrogel

“The Ca-alginate/polyacrylamide hydrogels were synthesized as described previously [29] with some modifications.”

Please specify the supplier for the raw materials used and explain in detail, what exactly is “modified” in difference to the cited procedure (ref. 29, 31-33) – in order to proof the novelty of the current study.

e.g. in difference to_ Li, G.; Zhang, G.P.; Sun, R.; Wong, C.P. Mechanical strengthened alginate/polyacrylamide hydrogel crosslinked by barium and ferric dual ions. J. Mater. Sci. 2017, 52, 8538-8545, doi:10.1007/s10853-017-1066-x

Answer: The information concerning supplier for the raw materials has been complemented, Please see the manuscript.

Again, the reviewer mentioned the “modification” procedure. Please see detailed explanation above. The authors thank the reviewer for the kind reminder and help.

Results and Discussion:

Line 89: 2.1. Bone Demineralization and Mechanical Tests

See above: In your introduction as state that “an optimized procedure has been developed” – so please refer to the existing procedure. Specify the novelty of the current study.

Answer: The authors felt confused about this comment, and did not do any corrections. If there is any further question, please feel free to let us know.

Conclusion:

Lines 359-367: “A braided PET tubular strut-reinforced Ca-alginate/polyacrylamide hydrogel vascular graft was fabricated with the assistance of a customized 3D casting mold. The mechanical properties concerning tensile strength, tensile elongation, dynamic circumferential compliance, compressive strength, and suture retention performance of the strut-reinforced grafts were improved significantly in comparison to native blood vessels and ePTFE prosthesis. Furthermore, the shape of strut-reinforced grafts could be maintained upon prolonged incubation in an aqueous environment. Thus, the strut-reinforced graft holds a great promise in meeting the required requisites for clinical operations. More importantly, it has a satisfying cytocompatibilityand hemocompatibility. The resent work paves the way to develop a new product-oriented small-diameter vascular graft.”

Please provide reference data to compare the cytocompatibility data of your grafts with comparable materials (also based on Ca-alginate/polyacrylamide) reported in literature – in order to support the statement of “satisfying cytocompatibility”.

Answer: With comparison to literatures data reported previously, the results show satisfying cytocompatibility of the devices. Please see Reference [31]. And one more reference was listed as follows.

[31] Darnell M C , Sun J Y , Mehta M , et al. Performance and biocompatibility of extremely tough alginate/polyacrylamide hydrogels[J]. Biomaterials, 2013, 34(33):8042-8048.

²  Guo P , Yuan Y , Chi F . Biomimetic alginate/polyacrylamide porous scaffold supports human mesenchymal stem cell proliferation and chondrogenesis[J]. Materials Science and Engineering: C, 2014, 42:622-628.

The authors thank the editor and both reviewers again for their diligent and careful work in reviewing the manuscript.

Sincerely,

Lu Wang, Ph.D.

Professor

Key Laboratory of Textile Science and Technology of Ministry of Education

College of Textiles, Donghua University, Shanghai, China

Round 2

Reviewer 1 Report

The authors have modified the paper according to reviewer's suggestion and properly clarified all the erosen issues. The work is thus suitable for publication in Polymer in its current form.